# Identification and Genetic Characterization of MERS-Related Coronavirus Isolated from Nathusius’ Pipistrelle (*Pipistrellus nathusii*) near Zvenigorod (Moscow Region, Russia)

**DOI:** 10.3390/ijerph20043702

**Published:** 2023-02-19

**Authors:** Anna S. Speranskaya, Ilia V. Artiushin, Andrei E. Samoilov, Elena V. Korneenko, Kirill V. Khabudaev, Elena N. Ilina, Alexander P. Yusefovich, Marina V. Safonova, Anna S. Dolgova, Anna S. Gladkikh, Vladimir G. Dedkov, Peter Daszak

**Affiliations:** 1Scientific Research Institute for Systems Biology and Medicine, Federal Service on Consumers’ Rights Protection and Human Well-Being Surveillance, 117246 Moscow, Russia; 2Central Research Institute for Epidemiology, Federal Service on Consumers’ Rights Protection and Human Well-Being Surveillance, 111123 Moscow, Russia; 3Biological Department, Lomonosov Moscow State University, 119234 Moscow, Russia; 4Saint-Petersburg Pasteur Institute, Federal Service on Consumers’ Rights Protection and Human Well-Being Surveillance, 197101 Saint-Petersburg, Russia; 5Department of Particularly Dangerous Diseases, Anti-Plague Center, Federal Service on Consumers’ Rights Protection and Human Well-Being Surveillance, 127490 Moscow, Russia; 6Martsinovsky Institute of Medical Parasitology, Tropical and Vector Borne Diseases, Sechenov First Moscow State Medical University, 119435 Moscow, Russia; 7EcoHealth Alliance, New York, NY 10018, USA

**Keywords:** bat-CoV, MERS-related coronaviruses, *Pipistrellus nathusii*, bats, hedgehogs, humans, camels, DPP4, spike protein, molecular docking

## Abstract

Being diverse and widely distributed globally, bats are a known reservoir of a series of emerging zoonotic viruses. We studied fecal viromes of twenty-six bats captured in 2015 in the Moscow Region and found 13 of 26 (50%) samples to be coronavirus positive. Of *P. nathusii* (the Nathusius’ pipistrelle), 3 of 6 samples were carriers of a novel MERS-related betacoronavirus. We sequenced and assembled the complete genome of this betacoronavirus and named it MOW-BatCoV strain 15-22. Whole genome phylogenetic analysis suggests that MOW-BatCoV/15-22 falls into a distinct subclade closely related to human and camel MERS-CoV. Unexpectedly, the phylogenetic analysis of the novel MOW-BatCoV/15-22 spike gene showed the closest similarity to CoVs from *Erinaceus europaeus* (European hedgehog). We suppose MOW-BatCoV could have arisen as a result of recombination between ancestral viruses of bats and hedgehogs. Molecular docking analysis of MOW-BatCoV/15-22 spike glycoprotein binding to DPP4 receptors of different mammals predicted the highest binding ability with DPP4 of the *Myotis brandtii* bat (docking score −320.15) and the *E. europaeus* (docking score –294.51). Hedgehogs are widely kept as pets and are commonly found in areas of human habitation. As this novel bat-CoV is likely capable of infecting hedgehogs, we suggest hedgehogs can act as intermediate hosts between bats and humans for other bat-CoVs.

## 1. Introduction

Coronaviruses (CoVs) have been responsible for three high impact outbreaks in the past two decades, including severe acute respiratory syndrome (SARS), the Middle East respiratory syndrome (MERS), and the coronavirus disease 2019 (COVID-19) pandemic. Each of these diseases affects the human respiratory system, causing a spectrum from asymptomatic or mild respiratory illness to severe pneumonia, acute respiratory failure, or death. The ongoing COVID-19 pandemic with more than 500 million confirmed cases, including more than six million deaths according to the WHO as of November 2022 [1], is caused by the SARS-CoV-2 coronavirus, which is most likely zoonotic. Which animal was the source of SARS-CoV-2 is not known, despite multiple reports of SARS-CoV-2 related viruses in various species of *Rhinolophus* bats (Asia) [2,3,4,5,6]. The intermediate hosts for SARS-CoV-2 are not precisely understood yet [7]. The earlier outbreak of SARS (also termed “atypical pneumonia”) was caused by the SARS-CoV coronavirus first emergent in 2002 in China. Due to high transmissivity, SARS-CoV rapidly spread, and one year later it had caused 8000 confirmed cases of infection in 29 countries (including European and North American countries) with 9.6–10% fatality [8,9,10]. SARS-CoV was introduced into the human population through carnivores (presumably civet or raccoon dog). Horseshoe bats are considered a reservoir host [8,10]. The outbreak of MERS was caused by the MERS-CoV coronavirus, transmitted to humans from infected dromedary camels [11]. MERS was first identified in Saudi Arabia in 2012 [12] and has now been reported in 27 countries leading to 858 known deaths due to the infection and related complications. The disease has a high fatality rate of up to 35% [13,14]. The origin of the virus is not fully understood yet, but phylogenetic analysis of different viral genomes suggests it originated in bats and passed to humans after circulating endemically in dromedary camels for around 30 years [15,16].

Bats represent around 1/5th of all mammalian biodiversity, featuring wide geographic distribution, long life spans, and they are also known to feed and roost near human communities. Bats are both known and putative reservoirs of several coronaviruses [17,18,19]. In addition to SARS-related, MERS-related and SARS-CoV-2-related viruses, bats also carry diverse coronaviruses which are not known to cause human diseases. However, some of these coronaviruses can bind to human cells in vitro, suggesting that bats are likely reservoirs of potential future zoonotic CoVs [20,21]. The aforementioned makes them an important subject of research. Identification of novel potential sources of infection among bats, especially among those near large cities, is important because this may help health authorities to: estimate emergence; and control wildlife or domestic animal reservoirs posing zoonotic risks.

Presently CoVs are divided into four genera: Alpha, Beta, Gamma and Delta. Betacoronavirus (β-CoV) is one of two genera of CoVs, which infect mammals, the second one is Alphacoronavirus (α-CoV) [20,22,23]. Presently, betacoronaviruses are classified into five subgenera: Embecovirus (clade A), Sarbecovirus (clade B), Merbecovirus (clade C), Nobecovirus (clade D), and Hibecovirus [24,25]. Of the viruses causing emergent diseases mentioned above, SARS and SARS-CoV-2 are members of the subgenera Sarbecovirus. MERS-CoV (which infects humans and camels) belongs to Merbecoviruses. MERS-related coronaviruses are closest relatives to MERS-CoVs; they include CoVs discovered in bats and hedgehogs [23,24,25,26]. MERS-related coronaviruses (MERSr-CoVs) have been reported from bats of South Africa (NeoCoV from *Neoromicia capensis*) [18,19,20], Mexico (Mex_CoV-9 from *Nyctinomops laticaudatus*), Uganda (MERSr-CoV PREDICT/PDF-2180 from *Pipistrellus* cf. *hesperidus*) [21], Netherlands (NL-VM314 from *Pipistrellus pipistrellus*) [22], Italy (BatCoV-Ita1 strain 206645-40 from *Hypsugo savii* and BatCoV-Ita2 strain 206645-63 from *Pipistrellus kuhlii*) [17], and China (BatCoV/SC2013 from *Vespertilio superans* [23], and multiple strains of HKU4- and HKU5-CoVs from *Tylonycteris* and *Pipistrellus* bats [27,28,29].

To date, few surveys for bat CoVs have been conducted in Russia and no MERSr-CoVs have been reported. In a recent work (2022), SARS-like coronaviruses circulating in a southern Russian region (near the Black Sea) were reported in local populations of horseshoe bats [30]. The main goal of this study is to survey bats in Central European Russia for CoVs and to genetically describe a novel MERS-related coronavirus from *Pipistrellus nathusii* (bat). In this paper, we show that MERS-related coronaviruses circulate not far from Moscow (a megacity and the Russian capital), while hypothesizing that hedgehogs may be susceptible to these viruses.

## 2. Materials and Methods

### 2.1. Sample Collection

In summer 2015, fecal samples were collected from 26 bats of the species *Myotis dasycneme* (*n* = 5), *Myotis daubentonii* (*n* = 5), *Myotis brandtii* (*n* = 3), *Nyctalus noctula* (*n* = 4), *Pipistrellus nathusii* (*n* = 6), *Plecotus auritus* (*n* = 2), and *Vespertilio murinus* (*n* = 1), inhabiting the Zvenigorodsky District of the Moscow Region (Sharapovskoe forestry, coordinates 55°41'24.0" N 36°42'00.0" E). No bats were killed during this study, and all bats were captured in mist nets and later released at the site of capture. Bat capture and sampling were conducted by professionally trained staff of the biology department of Lomonosov Moscow State University. After capture, fecal samples, rectal swabs, and ectoparasites were collected, while species, sex, reproductive and health status were usually determined by trained field biologists. Swab samples were kept in a transport media for transportation and stored with mucolytic agent (AmpliSens, Russia) at 4 °C during transportation to the laboratory. They were then stored at −80 °C before processing.

### 2.2. RNA Extraction and Reverse Transcription

RNA was extracted from bat fecal samples using the QIAamp Viral RNA Mini Kit (Qiagen, Germany). RNA carrier was dissolved in AVE buffer and added to AVL buffer according to manufacturer’s recommendations before extraction. 140 μL of fecal sample was added to the prepared AVL buffer with carrier RNA–Buffer AVE. Further steps were performed according to the original protocol. RNA was eluted with 60 μL of the AVE buffer and stored at −70 °C until evaluation. 10 μL of RNA was used for reverse transcription using Reverta-L reagents (AmpliSens, Russia). Second strand cDNA was obtained using the NEBNext Ultra II Non-Directional RNA Second Strand Module (E6111L, New England Biolabs). In order to increase input concentration, 24 μL of first strand product was added to 10 μL of H2O (milliQ) for subsequent steps.

### 2.3. PCR and NGS Screening of Faecal Samples for CoV RNA

PCR-screening for CoV RNA was performed using primers targeting alpha- and betacoronavirus species: 5′-CTTATGGGTTGGGATTATCC (CoV2A-F) and 5′-TTATAACAGACAACGCCATCATC (CoV2A-R) as described [28,29,30]. This generated ~400–500-bp amplicons from the RNA-dependent RNA polymerase (*RdRp*) gene. The following thermal cycling parameters were used: 94 °C for 3 min; followed by 10 cycles of 94 °C for 20 s, 55 °C to 45 °C (–1 °C per cycle) for 20 s, and 72 °C for 30 s; and then 42 cycles of 94 °C for 20 s, 45 °C for 20 s, and 72 °C for 30 s; and finally, 72 °C for 3 min [31]. PCR amplification products were analyzed by agarose gel electrophoresis. Positive PCR products were purified using AMPure beads and prepared for high throughput sequencing using the TruSeq protocol for Illumina. Sequencing was performed using Illumina MiSeq system to generate 250-bp paired-end reads. Reads were subjected to analysis via the following pipeline. Reads were filtered using Trimmomatic [32]. Then sequences of PCR primers, as well as simple repeats, were masked. Filtered reads with an unmasked region greater than 30 bp were collected and used for taxonomic analysis by comparing assembled contigs or individual reads to the NCBI non-redundant nucleotide and protein sequence databases using the blastn, blastx or tblastn algorithms as described [31].

### 2.4. Library Preparation and High-Throughput Viral Genome Sequencing

Double stranded cDNA was used for library preparation using the NEBNext Ultra II DNA Library Prep Kit for Illumina (New England Biolabs, Hitchin, UK). End preparation was performed according to the manufacturer’s protocol. For the adaptor ligation step, we chose Y-shaped adaptors compatible with Nextera XT Index Kit in the amount of 4 pM per reaction. PCR amplification with index adaptors in the amount of 7.5 pM per reaction was performed with Nextera XT Index Kit (Illumina, San Diego, CA, USA) in 25 μL total volume according to the NEBNext Ultra II DNA Library Prep Kit for Illumina protocol with 15 cycles.

High throughput sequencing was performed using Illumina HiSeq 1500 with the HiSeq PE Rapid Cluster Kit v2 and HiSeq Rapid SBS Kit v2 (500 cycles). Paired reads were filtered with Trimmomatic [32] using parameters SLIDINGWINDOW:4:25 MINLEN:40. After trimming, genome assembly and selection of *Coronaviridae* sequences, we obtained two contigs with lengths of 20,098 and 10,135 bp. Genome assembly was completed by SPAdes 3.15.0 [33]. Coronaviridae sequences were selected by BLASTn [34] of assembled contigs using all of the available *Coronaviridae* genomes as a reference. Read mapping was performed using bowtie2 [35].

Alignments of contigs to the closest MERS-CoV genomes (MG596802.1) revealed an uncovered 62 bp fragment between these contigs. Gaps within the assembled genome were closed and confirmed using Sanger sequencing. We performed Sanger sequencing of this area to connect two contigs and obtain full-genome sequencing with following primers: 1-forward ACATACGTGACAATGGTTCATTAG, and 1-reverse CTGTTGACTCTCTATAAATATAGAAC. Genome annotation was performed by Geneious 7.1.9 and edited manually. TRS-L and TRS-B alignment was made by Vector NTI software.

### 2.5. Phylogenetic Analysis

We downloaded beta-CoV sequences from GenBank using the keyword “Merbecovirus’’ as the primary filter to identify beta-CoVs from bats, humans, camels, and other mammals. We included 9 full genomes of CoVs from bats, 1 from the lama, 232 from camels, and 254 from humans. For phylogenetic tree construction, we only used complete genomes of viruses from highly represented hosts: individual trees were previously constructed separately for full genomes of camel and human *Merbecoviruses*. Clusters with *p*-distance > 0.001 were collapsed, and one sequence per cluster was selected randomly. The complete genome of the newly discovered CoV was aligned with all sequences using MAFFT v7 via the online service of RIMD (Research Institute for Microbial Disease of Osaka University, Osaka, Japan): https://mafft.cbrc.jp/alignment/software (accessed on 7 February 2023), and base pairs on 5′-end and 3′-end were then trimmed.

For phylogenetic analysis of genes encoding nucleocapsid protein, the partial sequences designated in metadata as “merbecovirus nucleocapsid gene” were downloaded from GenBank and combined with sequences extracted from complete genomes (sampling described above). The 128 obtained sequences were aligned using MAFFT v7. Approximately 110 bp from the 5′-end and ~200 bp from the 3′-end were trimmed. Finally, the phylogenetic tree was constructed on alignment of ~1308 bp, with the Best-fit model according to BIC: TIM2 + F + I + G4.

For phylogenetic analysis of the RdRp-encoding region as well as spike genes, a similar method of sampling was used. Specifically, the partial sequences designated in metadata as “merbecovirus RdRp” or “merbecovirus spike gene” were downloaded from GenBank, (2) combined with relevant sequences extracted from complete genomes, and aligned using MAFFT v7, with trimming at the 5′ and 3′-ends. For the RdRp-encoding region, a ~2801 bp alignment of 118 sequences was used for phylogenetic tree construction, with the best-fit model according to BIC: GTR + F + I + G4. For the Spike gene, a ~4026 bp alignment of 123 sequences was used for phylogenetic tree construction, with the best-fit model according to BIC was GTR + F + I + G4.

Phylogenetic analyses were performed using W-IQ-TREE (http://iqtree.cibiv.univie.ac.at/ (accessed on 7 February 2023)) with ModelFinder [36], tree reconstruction [37], and ultrafast bootstrap (1000 replicates) [38]. Phylogenetic trees and coevolutionary events were visualized using the online website (https://itol.embl.de/ (accessed on 7 February 2023)) with iTOL software [39].

### 2.6. Structural Modeling and Molecular Docking

The SWISS-MODEL server [40] was used to determine the three-dimensional structure of the MOW-BatCoV spike protein as well as the structures of DPP4 for *Myotis brandtii* [EPQ03439.1], *Pipistrellus kuhlii* [KAF6353216.1], *Erinaceus europaeus* [XP_016043930.1], *Felis catus* [NP_001009838.1], and *Mus musculus* [NP_034204.1]. The DPP4 sequence of *E. europaeus* contained X in the sequence, which we replaced with the W sequence. The 5T4E crystal structure of DPP4 of *Homo sapiens* was obtained from the RCSB.

The constructed models and the crystal structure of DPP4s of the studied organisms were docked to the modeled structure of the MOW-BatCoV Spike protein using HDOCK server [41]. The RBD (360–610 a.r.) of MOW-BatCoV and all DPP4 were used as docking sites. The docking results were analyzed in the PyMOL Molecular Graphics System, Version 2.0 [42].

## 3. Results

### 3.1. General Description

We collected and analyzed 26 fecal samples from six bat species: *M. dasycneme*, *M. daubentonii*, *M. brandtii*, *N. noctula*, *P. nathusii*, and *P. auritus*. All bats were visually healthy. Ectoparasite analysis yielded mites in 21/26 samples with 2 samples containing both mites and fleas. PCR of 420 bp fragments of *RdRp* gene yielded results in 13 of 26 fecal samples, giving an overall detection rate of 50%. Using Illumina MiSeq high-throughput sequencing of a 418–420 bp fragments of *RdRp* and data analysis as described [31], we confirmed the presence of CoVs. Five of the six species of bats sampled were infected by different CoV strains (*M. dasycneme*, *M. brandtii*, *M. daubentonii*, *N. noctula*, *P. nathusii*). Only one sample was negative for CoVs (from *P. auritus*). Of the investigated species, only *P. nathusii* were carriers of β-CoVs.

The results of a BLAST search for *RdRp* gene fragments from *P. nathusii* in which coronavirus infection has been found showed in Table 1. Four of six *P. nathusii* were carriers of α-CoVs. The *RdRp* of α-CoVs shared >98% sequence identity to EU375864.1 (P.nat/Germany/D5.16/2007) or >99.5% sequence identity to EU375869.1 (P.nat/Germany/D5.73/2007) which were found in *P. nathusii* from Germany in 2007 [43].

Three of six *P. nathusii* were carriers of β-CoVs. All *RdRp* gene fragments of β-CoVs shared >98% sequence identity to KC243390.1 (BtCoV/8-724/Pip_pyg/ROU/2009) which was previously found in *Pipistrellus pygmaeus* from Romania and published in 2013 [44]. Topology of the phylogenetic tree based on the *RdRp* gene fragments revealed that the bats were infected with the same virus, it was named MOW-BatCoV. We presumed that these represent a potentially novel betacoronavirus in *P. nathusii* and sequenced the complete viral genome of sample №22.

*P. nathusii* is widely distributed across Europe. Figure 1A shows a bat caught during the process of fecal sample collection. Metagenome sequencing of total RNA extracted from sample №22 resulted in 248.8 million paired reads (SRR15508267); 0.01% of them were mapped to initially obtained *Coronaviridae* contigs resulting in the complete genome. It was named MOW-BatCoV strain 15-22 (Bat-CoV/P.nathusii/Russia/MOW15-22/2015, accession numbers ON325306). *RdRp* gene fragments of viral genomes from two other samples (№16, №33) have been deposited in GenBank as Bat-CoV/P.nathusii/Russia/MOW15-16/1/15 (acc. number ON676527) and Bat-CoV/P.nathusii/Russia/MOW15-33/1/15 (acc. number ON676528).

Bat-CoV/P.nathusii/Russia/MOW15-22/2015 contains 30,257 bases, with 43.72% G + C content. The genomic organization of the virus is similar to that of other members of *Merbecovirus*, namely: ORF1ab encoding putative mature nonstructural proteins, including RdRp (RNA-dependent RNA polymerase); S (spike protein); the genes encoding nonstructural proteins NSP3, NSP4a, NSP4b and NSP5; E (envelope protein); M (membrane glycoprotein); N (nucleocapsid phosphoprotein); the gene encoding nonstructural protein NSP8 (Figure 2). Predicted proteins and transcription regulatory sequences are summarized in Table 2.

The size and genomic localization of the nonstructural proteins (NSP 1–16) encoded by ORF1ab were predicted by sequence comparison with MERS-CoV (human HCoV-EMC/2012) and other beta-CoV species. The nonstructural proteins characteristics and 15 expected cleavage sites are shown in Table 3. In ORF1ab, the sequence is “UUUAAAC”, which is conserved throughout all CoVs and located at 13,497–13,503 nucleotide position. A predicted leader transcription regulatory sequence (TRS-L), as well as seven putative transcription regulatory TRS-B group sequences, representing signals for the discontinuous transcription of subgenomic mRNAs (sgmRNAs), have been identified (Table 3). All TRS have a conserved AACGAA motif forming the conserved TRS core in β-CoVs [46], with only one G/A modification in ORF3 (Figure 3).

The ICTV proposed that viruses sharing >90% amino acid sequence identity in the conserved replicase domains should be considered conspecific. A separate comparison of the amino acid sequences of seven conserved ORF1ab domains (as suggested by the ICTV) for formal CoV species delineation was made, and only the NSP3 (ADRP) amino acid sequence is below the 90% threshold value in comparison with MERS. ORF1ab possessed 81.5–82.3% n.a. identities to the ORF1ab of other members of *Merbecovirus*. Comparison of the seven conserved domains in replicase polyprotein pp1ab with other coronaviruses showed that MOW-BatCoV/15-22 possessed a.a. identity to other members of *Merbecovirus* as follows: NSP3 (ADRP)—68.7%; NSP5 (3CLpro)—90.8%; NSP12 (RdRp)—96.6%; NSP13 (Hel)—97.8%; NSP14 (ExoN)—97.9%; NSP15 (NendoU)—93.3%; and NSP16 (O-MT)—93.7%.

MOW-BatCoV/15-22 MERSr-CoV has nucleic acid identity from 81.32% to 82.46% (with coverage 82–85%) to the ten closest MERS or MERSr-CoVs from bats (*Vespertilio sinensis*, *Vespertilio superans*, *Pipistrellus* cf. *Hesperidus*), humans, and camels reported between 2013 and 2015 (Appendix A). The highest sequence identity was to two MERSr-CoVs from bats (*Hypsugo savii*, *Pipistrellus kuhlii*) caught in Italy in 2011.

### 3.2. Phylogenetic Analysis

Our phylogenetic analysis of complete MERSr-CoV genomes showed distinct clades (Figure 3): (I) consists of nine CoV sequences from hedgehogs; (II) consists of 17 CoV sequences from bats; (III) consists of sequences belonging to two described subclades [47]. These subclades are: (a) CoVs from bats, including novel MOW-BatCoV/15-22 virus; and (b) CoVs from humans and camels, as well as one from a bat (Neoromicia/5038, collected in South Africa in April of 2015 from the bat *N. capensis*).

In the N-tree MOW-BatCoV/15-22 together with three other MERSr-CoVs from bats (namely *Neoromicia/5038*, *Bat-CoV/H.sav/Italy/206645-40/2011* and *Bat-CoV/P.khulii/Italy/206645-63/2011*) forms a distinct subclade closely related to human and camel MERS-CoVs (Figure 4a).

In the RdRp-tree, MOW-BatCoV/15-22 formed a subclade MOW-BatCoV/15-22 along with the only Neoromicia/5038 and human/camel viruses, but MERS-related coronaviruses from *H. savii* and *P. kuhlii* (from Italy) fell into another clade (Figure 4b).

Phylogenetic analysis of spike genes shows that MOW-BatCoV/15-22 is most closely related to Neoromicia/5038. Unexpectedly, both MOW-BatCoV/15-22 and Neoromicia/5038 showed the closest relationship to CoVs from hedgehogs, forming a distinct branch among *Merbecovirus* (Figure 4c).

### 3.3. Docking

To predict and analyze the interaction of MOW-BatCoV/15-22 spike glycoprotein with DPP4 receptors of different mammals, three-dimensional structures of these proteins were obtained by homologous modeling. DPP4 proteins of two bats (*Myotis brandtii*, *Pipistrellus kuhlii*), a hedgehog (*Erinaceus europaeus*), a domestic cat (*Felix catus*), and a mouse (*Mus musculus*) were used for analysis.

MOW-BatCoV/15-22 spike protein structure was determined using the 6Q04 reference structure of human MERS-CoV spike protein. We selected the protein structure of the amino acid sequence with the highest homology (61.41%) to the input sequence as a template. The resulting model had a GMQE of 0.61.

For the DPP4 structure models of mammals, the reference human FAP alpha was used. The best 6Y0F structure models were determined for *M. brandtii* and *P. kuhlii* DPP4 with 92.14% identity (GMQE = 0.91) and 92.31% identity (GMQE = 0.93) respectively. The best reference structure for DPP4 structure models of *E. europaeus*, *F. catus*, and *M. musculus* was 2QT9 with identities of 84.97% (GMQE = 0.9), 87.97% (GMQE = 0.89) and 84.15% (GMQE = 0.9) respectively.

Molecular docking allowed us to determine the best binding cluster for spike protein of MOW-BatCoV/15-22 and DPP4 proteins of the organisms studied. The highest binding was predicted between MOW-BatCoV/15-22 spike protein and DPP4 of *M. brandtii* (docking score −320.15). The second highest binding was predicted for *E. europaeus* (docking score −294.51), which is consistent with phylogenetic analyses. Docking results predicted higher binding of the MOW-BatCoV/15-22 spike to *H. sapiens* DPP4 (docking score −290.79) than to *P. kuhlii* DPP4 (−274.21), *M. musculus* (docking score −262.74) or *F. catus* (docking score −248.18).

Molecular docking showed that spike-protein of MOW-BatCoV/15-22 and DPP4 of *M. brandtii* and *E. europaeus* share binding sites. Thus, 43 binding sites were shown for *M. brandtii* and 42 for *E. europaeus* in the spike-protein, including 31 identical sites (Table 4). We suppose that since there are many overlapping binding sites between spike-protein of MOW-BatCoV/15-22 and DPP4 of *M. brandtii* and *E. europaeus*, MOW-BatCoV/15-22 likely can infect both bats and hedgehogs.

## 4. Discussion

### 4.1. Abundance of Coronaviruses in Central European Russia

A large number of β-CoVs has been identified from bats globally. In our study, we analyzed twenty-six bats of six different species widely distributed in the central European part of the Russian Federation. A high percentage of studied bats (50%) was positive for alpha- or betacoronaviruses (or both). This observation corresponds to previous reports about coronavirus distribution in bats: the proportion of bats with detectable coronaviral RNA in feces, or in fecal or oral swabs, could reach up to 50–100% depending on the season, the geographic location of bats, and bat species [29]. Last year, two novel SARS-like coronaviruses (Khosta-1, Khosta-2) from greater and lesser horseshoe bats were described in southern regions of Russia. The authors who described these viruses showed that 14% and 1.75% of greater horseshoe bats were positive for Khosta-1 and Khosta-2 virus, respectively (up to 62.5% of Khosta-1 positive in some caves) [48]. It can be supposed the variability of coronavirus species may be even higher in southern regions of Russia. Meanwhile, there is no information about bat coronaviruses in Central Russia. Actually, this investigation is the first report about the abundance and variability of coronaviruses in bats which inhabit the Central European part of Russia near Moscow (a megacity and the capital). It is self-evident that seasonal fluctuations of coronaviruses in this region are very important and need further careful study.

We sequenced and assembled the complete genome of the most important novel coronavirus: the MERS-related betacoronavirus MOW-BatCoV strain 15-22. This coronavirus was found only in one of six investigated bat species, namely *P. nathusii* (common name—Nathusius’ pipistrelle). Out of six *P. nathusii* samples, three (50%) were carriers of MOW-BatCoVs. The animals were caught at the same time in the same geographic location. Therefore, we believe they belonged to the same colony and had close physical contact while roosting.

### 4.2. Taxonomic Position of the MOW-BatCoV: Whether It Is a New Species

According to phylogenetic analysis of complete genomes, MOW-BatCoV/15-22, together with a few MERS-related bat viruses, falls into the clade of human/camel MERS viruses. MOW-BatCoV and BtCoVNeo5038/KZN/RSA/2015 together are closest to MERS-CoV. The replicase polyprotein (*RdRp*) gene of the new virus showed more than 90% homology to sequences of the other members of *Merbecoviruses* for six of seven NSP domains. Seventh domain, the NSP3 (ADRP) showed 68.7% homology. According to demarcation criteria of ICTV [30], we believe *MOW-BatCoV* represents the same species of *Merbecovirus* as BtCoVNeo5038/KZN/RSA/2015 of a *N*. *capensis* collected in South Africa (KwaZulu-Natal province) [49,50].

### 4.3. Distribution of MERS-Related Viruses around Europe

The complete genome of MOW-BatCoV/15-22 showed the highest similarity to MERS-related viruses which have been reported in Italy, in bats (*Hypsugo savii*, *Pipistrellus khulii*) captured in 2011 [26]. The Moscow Region of Russia and Italy are far from each other (Figure 1B, the countries with related MERS-related viruses are marked in red). However, bat migration routes can explain the similarity between viruses found in bats from Italy and Russia. Moreover, phylogenetically *H. savii* and *P. khulii* are close relatives with *P. natusii*. Thus, the discovery of closely related species of viruses in closely related species of bats can be considered as indirect evidence of the migration of MERS-related viruses across Europe. *P. nathusii* is a migratory bat inhabiting the most of Europe, from Fennoscandia and the British Isles in the north to the Mediterranean in the south. The breeding areas of this species are regions of north-eastern Europe. Due to a lack of aerial insects in winter, Nathusius’ pipistrelles from Central European (Germany, Poland) and northeastern populations (Fennoscandia, the Baltic countries, Belarus, Russia) carry out long-distance flights migrating in the late summer (over approximately two months with stopping for mating) to Switzerland, the Benelux countries, France, Spain, Italy, and Croatia [26,51]. Before 2022, the longest documented migration record of this species was 2224 km, between southern Latvia and northern Spain [51]. In 2022 it was reported Nathusius’ pipistrelle bats set a new bat migration record with a 2486 km flight from Russia to French Alps [52]

While migrating, *P. nathusii* may come into contact with bats of the same species when mating. They also can contact bats of other species in roosting areas. Research on European migration routes of pipistrelle bats could help us to understand the ways in which coronaviruses spread across Europe with migratory bat species.

### 4.4. Hedgehogs as Potential Intermediate Hosts between Bats and Other Animals

According to phylogenetic analysis of complete genomic sequences as well as *N* and *RdRp* sequences, ten coronaviruses from bats caught in different geographic regions of Earth (including the novel MOW-BatCoV/15-22) form a common phylogenetic clade with MERS-CoVs isolated from humans and camels. However, phylogenetic analysis of the spike encoding genes demonstrated similarity of two bat coronaviruses (the novel MOW-BatCoV, NeoCoV) to coronaviruses isolated from *Erinaceus europaeus* (the European hedgehog).

Recently MERS-related viruses were discovered in hedgehogs (*Erinaceous*). Hedgehog carriers of betacoronaviruses were found in China [53], England [25], Germany [54], France [55], and Poland [56]. Phylogenetic analysis carried out previously showed that betacoronaviruses from Chinese and German hedgehogs (Ea-HedCoV HKU31, Beta-CoV Erinaceus/VMC/DEU/2012) are closely related to NeoCoV and BatCoV from African bats in the spike region. Therefore, the authors suggested that the bat viruses arose as a result of recombination between hedgehogs and bat viruses [25]. Our independent discovery of another virus from the European bat *P. nathusii* (which appears to be closely related to viruses from hedgehogs in the *spike*, but not *N* and *RdRp*, genes) supports the idea of recombination between ancestral viruses of bats and hedgehogs. The overlap of *P. nathusii* and *E. europaeus* habitats raises the possibility that this recombination represents ancestral interspecies transmission. Our findings support the need for wider surveillance of MERSr-CoVs in both bats and hedgehogs.

The similarity between spike genes of viruses from bats and hedgehogs living in the same geographic region (namely modern Europe) raises the possibility of interspecies transmission at the present time. For instance, Middle East syndrome coronavirus (MERS-CoV) could have originated in bats and passed to humans through dromedary camels. It was previously suggested that MERS-CoV originated from an ancestral virus in a bat reservoir and spilled over into dromedary camels around 400 years ago, where it circulated endemically before emerging in humans in 2012 [15,16]. Presently, camels play an important role as a constant reservoir of MERS-CoVs and transmit viruses to people [25,53,54], while bats are widely considered to be “the evolutionary, disposable source of the virus” [57]. However, besides dromedaries which are the proven source of human MERS there could be another intermediate host among animals in the wildlife.

### 4.5. MOW-BatCoV/15-22 Possibility to Infect Other Mammalian Species

Of known bat viruses, MOW-BatCoV/15-22 from *P. nathusii* (European Russia) and NeoCoV from *N. capensis* (S. Africa) are the closest to Middle East respiratory syndrome coronavirus (MERS-CoV) which could infect humans and dromedary camels [57,58]. The coronavirus cell tropism and ability to infect hosts is determined primarily by the receptor binding domain (RBD) of the spike protein.

Phylogenetic analysis of complete genomic sequences, as well as *N* and *RdRp* sequences, suggest that ten viruses from bats found in distinct geographic regions (MOW-BatCoV/15-22 from Russia, MG596802.1 and MG596803.1 from Italy, NeoCoV Neoromicia/5038 from South Africa, and multiple strains from bats in China—MG021452.1, MG021451.1, MG987420.1, MG987421.1, KX442565.1, KX442564.1) form a distinct phylogenetic clade with MERS-CoVs from humans and camels, with high bootstrap support. However, phylogenetic analysis of spike protein encoding genes demonstrated similarity of two of these ten bat viruses (the novel MOW-BatCoV and NeoCoV) to CoVs from *Erinaceus europaeus* (the European hedgehog) [15,55,56]. Unlike SARS-CoV and SARS-CoV-2, which bind to angiotensin-converting enzyme 2 (ACE2), MERS-CoV targets the cell surface receptor dipeptidyl peptidase 4 (DPP4, also known as CD26) [59,60]. DPP4 is relatively conserved among mammals. Therefore MERS-CoV is capable of infecting a wide range of cell lines derived from humans, non-human primates, bats, swine, horses, rabbits, civets, and camels, but not from mice, hamsters, dogs, ferrets, or cats [19,61]. The HKU4 *Merbecoviruses* from Chinese bats are similar to MERS-CoV in spike protein genes and can use the MERS-CoV receptor DPP4. MERSr-CoVs HKU4 from Chinese bats have RBDs that can bind to human DPP4 with low affinity [59], suggesting potential to infect humans and adapt to more efficient cell entry [26]. Another MERS-like coronavirus, Hp-BatCoV HKU25 (from Chinese pipistrelle bats), can bind DPP4 for entry to DPP4-expressing cells, although with lower efficiency than that of MERS and HKU4 viruses [62]. At least some Merbecoviruses can bind the ACE2 receptor (namely, Bat-CoV-PREDICT/PDF-2180 and NeoCoV) [63]. These facts raise questions: if MOW-BatCoV/15-22 can infect humans or other animals; and if it can use the DPP4 receptor or not. We found that the amino acid composition of the RBD domain of the MOW-BatCoV/15-22 virus differs in the same degree from the RBD domains of those viruses that interact with both DPP4 receptors (32–36.6% a.a. similarity) and ACE2 receptors (33–33.7% a.a. similarity). Although we cannot rule out that MOW-BatCoV/15-22 binds to other cell receptors (e.g., ACE2), it is more likely it binds to DPP4. It is possible that the RBD domain of MOW-BatCoV/15-22 interacts with DPP4 across amino acids 366–624 within the S1 subunit.

We believe that only a small number of potential mutations separates MOW-BatCoV/15-22 from being able to infect humans. In a previously published work, it was shown that HKU4 coronavirus became infectious for human cells after two mutations in the spike gene [64]. There are two important amino acid motifs, namely hPPC (recognized by furin proprotein convertase) and hECP (recognized by endosomal cysteine protease Cathepsin L). In *MERS-CoV*, which causes Middle East respiratory syndrome, the hPPC motif is Arg748-Ser749-Val750-Arg751-Ser760, and the hECP motif is Ala763-Phe764-Asn765. In original, non-infectious for humans HKU4 virus, the hPPC motif is Ser746-Thr747-Phe748-Arg749-Ser750, and the hECP motif is Asn762-Tyr763-Thr764. When Ser746 in *HKU4* was changed to Arg746 (to make motifs recognizable by protease), and Asn762 became Ala762 (to destroy a potentially existing N-linked glycosylation site), the virus acquired the ability to enter into human cells.

Two variants of the hPPC motif can be predicted for MOW-BatCoV/15-22: Pro758-His759-Ser760-Arg761 (based on comparison with MERS and HKU4, which interact with DPP4 receptor) or Ser760-Arg761-Thr762-Asn763 (comparison with SARS-CoV-2 which interacts with the ACE2 receptor). It is possible that substitution of either Pro758 (in the first predicted hPPC motif) or Asn763 (in the second predicted hPPC motif) can lead to a furin cleavage site formation and its subsequent recognition by furin, which can increase the ability to infect human cells.

The hEPC motif in MOW-BatCoV/15-22 virus is Ala772-Tyr773-Pro774, and it is similar to hEPC of other coronaviruses. In this motif, as in HKU4, there are no amino acids containing nitrogen, which removes the possibility of an N-linked glycosylation site, and there is Tyr773, which is a conserved aa in coronaviruses. Therefore, the virus can employ this motif for breaking into human cells without any additional mutations (at least, in theory). Thus, theoretically, a sole mutation in MOW-BatCoV/15-22 virus spike protein (namely in the hPPC motif) could be enough to enable the virus to infect human cells.

We used another calculation method to estimate the probability of the virus infecting humans or animals who often contact humans, computational molecular docking analysis. Modeling identified the DPP4 receptors of species that MOW-BatCoV/15-22 spike glycoprotein could bind to. The lowest protein-protein binding was predicted in the interaction of MOW-BatCoV/15-22 spike protein and DPP4 of the mouse and cat. This finding is supported by the fact that MERS-CoV cannot infect cell lines derived from mice or cats [19,61]. The highest protein-protein binding was predicted in the interaction of MOW-BatCoV/15-22 spike protein and DPP4 of the bat *M. brandtii*, then of the hedgehog, *E. europaeus*. This is consistent with phylogenetic analyses and may indicate evolutionary relationships and exchange of spike genes (as result of recombination) between ancestral MERSr-CoVs of bats and hedgehogs. These data, as well as previous reports on wide distribution of MERSr-CoVs in hedgehogs, suggest these mammals can be a natural reservoir of this clade of novel betacoronaviruses, for example subgenus *Merbecovirus* [25,54,55,56]. They also suggest that the pathway of emergence of MERS-CoV may be more complicated than currently thought. Additionally, hedgehogs are increasingly kept as pets across the globe, with substantial numbers bred and shipped internationally, including the Americas (where MERSr-CoVs have not yet been reported). We suggest that hedgehogs should be considered, tentatively, as potential intermediate hosts for spillover of MERSr-CoVs between bats and humans, and that screening of captive hedgehogs should be conducted to rule out potential for future zoonotic spillover.

## 5. Conclusions

+In conclusion, our results show that the MERS-related betacoronaviruses (namely MOW-BatCoV) are circulating among the *Pipistrellus nathusii* (bat) population in Central European Russia (near Moscow). Further studies are needed to explore the distribution of MERS-related coronaviruses among bats in Europe. Molecular docking analysis allowed us to estimate the interaction between MOW-BatCoV spike protein and DPP4 proteins of bats and hedgehogs. We assert that MOW-BatCoV likely has the ability to infect hedgehogs. Further studies are needed to explore the potential transmission of coronaviruses between bats and hedgehogs.

## Figures and Tables

**Figure 1 ijerph-20-03702-f001:**
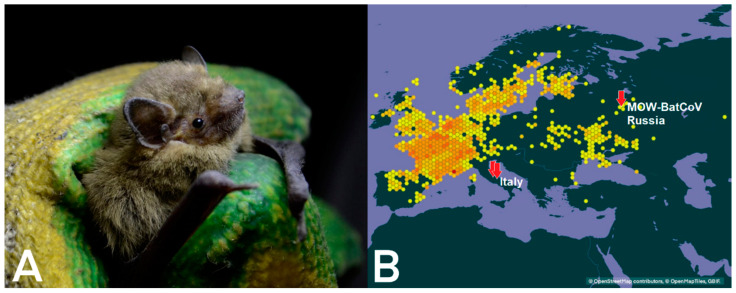
(**A**) *P. nathusii*, also known as Nathusius’ pipistrelle, studied here; (**B**) geographical distribution of *P. nathusii* in Europe. The dark-yellow spots represents its habitat in accordance with data present on web-site devoted to zoology [45]. The red arrows represents the location in Europe where different MERS-related Bat-CoVs were found.

**Figure 2 ijerph-20-03702-f002:**
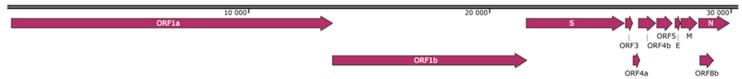
Genomic organization of MOW-BatCoV, strain 15-22. Illustrated reading frames (ORFs) have the following order and encode the listed genes: ORF1ab (putative mature nonstructural proteins, including RNA-dependent RNA polymerase (RdRp); S (Spike); ORF3 (hypothetical NS3); ORF4a (hypothetical NS4a); ORF4b (hypothetical NS4b); ORF5 (hypothetical NS5); E (envelope); M (membrane glycoprotein); N (nucleocapsid phosphoprotein); and ORF8b (hypothetical NS8b).

**Figure 3 ijerph-20-03702-f003:**
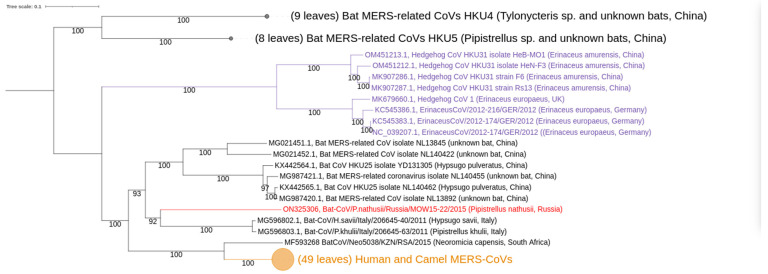
Phylogenetic trees of *Merbecovirus* genomes (from complete genomes only). The phylogenetic tree was constructed from 84 complete genome sequences, excluding 5′- and 3′-ends (29,757–30,331 bp). Numbers show bootstrap values. Best-fit model of substitution according to BIC: GTR + F + I + G4. The virus described in this study is labeled in red bold.

**Figure 4 ijerph-20-03702-f004:**
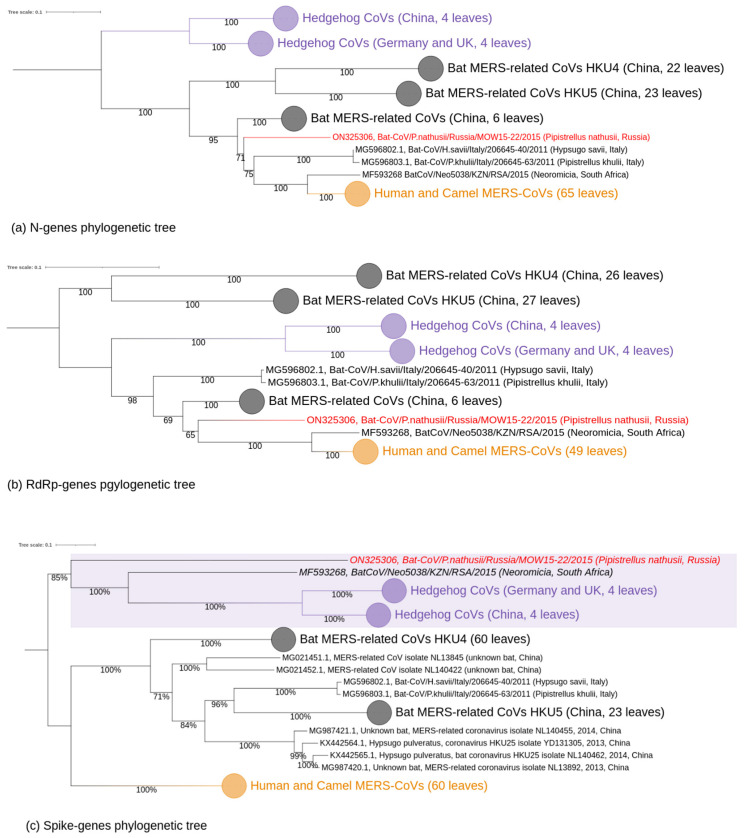
Phylogenetic trees of protein coding regions of *Merbecovirus* genomes. Numbers show bootstrap values. The virus described in this study is labeled in red. (**a**) tree constructed on 65 partial N-gene sequences (1272–1297 bp). Best-fit model of substitution according to BIC: TIM2 + F + I + G4; (**b**) tree based on 2798–2802 of partial RdRp (NSP12) protein coding regions. Best-fit model of substitution according to BIC: GTR + F + I + G4. (**c**) tree based on the partial spike glycoprotein protein encoding regions. Best-fit model of substitution according to BIC: GTR + F + I + G4.

**Table 1 ijerph-20-03702-t001:** *P. nathusii* in which natural coronavirus infection was investigated. Samples where β-CoVs were detected are marked in bold.

Sample ID	Gender	Age	Ectoparasites	PCR Product	Fragment Length, bp	GenBank ID, Nearest	Identity, %	Genus
**Bat№16**	**F**	**Semi-adult**	**Mites**	**+**	**418**	**KC243390.1**	**98.35**	**Beta**
					420	EU375869.1	99.51	Alpha
Bat№21	F	Semi-adult	Mites	+	420	EU375864.1	98.77	Alpha
**Bat№22**	**M**	**Semi-adult**	**Mites**	**+**	**418**	**KC243390.1**	**98.51**	**Beta**
Bat№23	M	Adult	Mites	+	420	EU375869.1	99.51	Alpha
Bat№25	F	Adult	Many Mites	-	-	-	-	-
**Bat№33**	**M**	**Semi-adult**	**Many Mites**	**+**	420	EU375864.1	98.77	Alpha
					**418**	**KC243390.1**	**98.35**	**Beta**

**Table 2 ijerph-20-03702-t002:** Genomic localization of predicted MOW-BatCoV/15-22 protein sequences.

ORF	Nt Position (Start-End)	No. of Amino Acids	Putative Leader TRS-L and TRS-B
ORF1a	175–13,524	4449	TRS-L
ORF1ab	175–21,587	7137	TRS-L
S prot	21,529–25,629	1367	TRS-B
ORF3	25,645–25,959	105	TRS-B
ORF4a	25,968–26,252	95	TRS-B
ORF4b	26,173–26,925	251	No
ORF 5	26,935–27,612	226	TRS-B
E prot	27,691–27,939	83	TRS-B
M prot	27,954–28,619	222	TRS-B
Nprot	28,675–29,982	436	TRS-B
ORF 8b	28,721–29,314	198	No

**Table 3 ijerph-20-03702-t003:** Proteins and 15 expected cleavage sites encoded by ORF1ab of MOW-BatCoV/15-22 MERS-related coronavirus. Superscript numbers indicate positions in polyprotein pp1a/pp1ab or position in available sequence with the supposition of a ribosomal frameshift based on the conserved slippery sequence (UUUAAAC) of Coronaviruses. Beginning at nucleotide position 13,497–13,503 are: ADRP—ADP-ribose 1-phosphatase, PL2pro—papain-like protease 2, 3CLpro—coronavirus NSP5 protease, Hel—helicase, NTPase—nucleoside triphosphatase, ExoN—exoribonuclease, NMT N7—methyltransferase, NendoU—endoribonuclease, and OMT—2′ O-methyltransferase.

NSP	Position of Putative Cleavage Sites	Protein Size (Number of Amino Acids)	Putative Functional Domain(s)
NSP1	Met^1^-Gly^195^	195	
NSP2	Asp^196^-Gly^858^	663	
NSP3	Ala^859^-Gly^2798^	1940	ADRP, PL2pro
NSP4	Ala^2799^-Gln^3305^	507	
NSP5	Ser^3306^-Gln^3611^	306	3CLpro
NSP6	Ser^3612^-Gln^3903^	292	
NSP7	Ser^3904^-Gln^3986^	83	
NSP8	Ala^3987^-Gln^4185^	199	Primase
NSP9	Asn^4186^-Gln^4295^	110	
NSP10	Ala^4296^-Gln^4435^	140	
NSP11	Ser^4436^-Ile^4449^	14	Short peptide at the end of ORF1a
NSP12	Val^4450^-Gln^5369^	920	RdRp
NSP13	Ala^5370^-Gln^5967^	598	HEL, NTPase
NSP14	Ser^5968^-Gln^6491^	524	ExoN, NMT
NSP15	Gly^6492^-Gln^6834^	343	NendoU
NSP16	Ala^6835^-Cys^7137^	303	OMT

**Table 4 ijerph-20-03702-t004:** Predicted MOW-BatCoV/15-22-DPP4 binding sites of *M. brandtii* and *E. europaeus*.

	*Myotis brandtii*	*Erinaceus europaeus*
binding sites	36, 163, 165, 166, 188, 189, 191, 230, 232, 233, 370, 389, 417, 419, 485, 486, 487, 488, 489, 490, 585, 611, 613, 617, 618, 619, 620, 637, 696, 697, 698, 701, 703, 704, 705, 707, 716, 718, 719, 720, 722, 724, 734	36, 38, 39, 154, 156, 157, 158, 159, 163, 165, 166, 188, 189, 190, 191, 197, 230, 232, 233, 370, 389, 416, 417, 419, 485, 486, 487, 488, 489, 490, 585, 613, 617, 618, 619, 620, 697, 698, 699, 703, 704, 718
overlapping binding sites	36, 163, 165, 166, 188, 189, 191, 230, 232, 233, 370, 389, 417, 419, 485, 486, 487, 488, 489, 490, 585, 613, 617, 618, 619, 620, 697, 698, 703, 704, 718	

## Data Availability

The raw data of metagenome sequencing of total RNA are deposited in the GenBank under the accession numbers ON325306. *RdRp* gene fragments sequences have been deposited in the GenBank under accession numbers ON676527 and ON676528. The first version of this paper has been published as a preprint [65].

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
