# Peer review of "Identification and Genetic Characterization of MERS-Related Coronavirus Isolated from Nathusius’ Pipistrelle (*Pipistrellus nathusii*) near Zvenigorod (Moscow Region, Russia)"

_ijerph, 2023, doi:10.3390/ijerph20043702_

Round 1
Reviewer 1 Report
Reference list is missing. Also, the figure resolution is too low to see the details. Please address these first, as we cannot peer review without them.
Author Response
Thank you for your comments.
- Reference list is checked.
- The figures of phylogenetic trees have now changed significantly. We hope the new version is much better.
Reviewer 2 Report
Journal: IJERPH (ISSN 1660-4601)
Manuscript ID: ijerph-2119089
Title: Identification and genetic characterization of MERS-related coronavirus isolated from Nathusius’ pipistrelle (Pipistrellus nathusii) near Zvenigorod (Moscow region, Russia)
The topic is of interest, and the manuscript is well illustrated.
Major Comments:
1. Are there controversies in this field? What are the most recent and important achievements in the field? In my opinion, answers to these questions should be emphasized. Perhaps, in some cases, novelty of the recent achievements should be highlighted by indicating the year of publication in the text of the manuscript.
2. The results and discussion section is very weak and no emphasis is given on the discussion of the results like why certain effects are coming in to existence and what could be the possible reason behind them?
3. Structural modeling and molecular docking: need more insights.
4. Results and conclusion: The section devoted to the explanation of the results suffers from the same problems revealed so far. Your storyline in the results section (and conclusion) is hard to follow. Moreover, the conclusions reached are really far from what one can infer from the empirical results.
5. The discussion should be rather organized around arguments avoiding simply describing details without providing much meaning. A real discussion should also link the findings of the study to theory and/or literature.
6. Spacing, punctuation marks, grammar, and spelling errors should be reviewed thoroughly. I found so many typos throughout the manuscript.
7. English is modest. Therefore, the authors need to improve their writing style. In addition, the whole manuscript needs to be checked by native English speakers.
8. Conclusions: need a solid storyline with authors own language.
Author Response
Thank you for your comments.
Question 1. Are there controversies in this field? What are the most recent and important achievements in the field? In my opinion, answers to these questions should be emphasized. Perhaps, in some cases, novelty of the recent achievements should be highlighted by indicating the year of publication in the text of the manuscript.
Answer1: Thank you for your comments, we have tried to make changes to the text of the manuscript. This area of scientific knowledge is developing very actively. It is very difficult to cover all the previosly published important works. We have removed some paragraphs from the introduction that were not directly related to the main topic of the article (the main topic of the article is a description of a new MERS-related virus from an area heavily populated by people in which no such viruses had been described before).
Question 2. The results and discussion section is very weak and no emphasis is given on the discussion of the results like why certain effects are coming in to existence and what could be the possible reason behind them?
Answer2: We have significantly changed and revised the text in the discussion section. The authors hope that the new version of the text is better understandable.
Question 3. Structural modeling and molecular docking: need more insights.
Answer 3: On the contrary, after careful consideration, we have decided that some of the information provided about Structural modeling is redundant. We have removed the figure depicting the protein-protein interaction because it is not informative enough. We plan to conduct experiments in the near future and evaluate the in vitro interaction between spike proteins and animal receptors.
Question 4. Results and conclusion: The section devoted to the explanation of the results suffers from the same problems revealed so far. Your storyline in the results section (and conclusion) is hard to follow. Moreover, the conclusions reached are really far from what one can infer from the empirical results.
Question 5. The discussion should be rather organized around arguments avoiding simply describing details without providing much meaning. A real discussion should also link the findings of the study to theory and/or literature.
Answer 4 and 5: Thank you for your comments, they really helped us to see the weaknesses of the text. The discussion section has been heavily modified. The "Conclusion" section has also been slightly changed.
Question 6. Spacing, punctuation marks, grammar, and spelling errors should be reviewed thoroughly. I found so many typos throughout the manuscript.
Question 7. English is modest. Therefore, the authors need to improve their writing style. In addition, the whole manuscript needs to be checked by native English speakers.
Answer 6 and 7: We've fixed languages bugs (corrected by a professional proofreader).
Reviewer 3 Report
This is a timely study that adds to the growing body of epidemiological data related to coronaviruses. It is simple yet elegant and of a scope that provides a critical snapshot without being overly ambitious. There is nothing novel about the approach and the findings are certainly not groundbreaking. However, it is the simplicity of the approach that adds to the credibility of the acquired data. Those data should be made broadly available to the community of coronavirus researchers for inclusion in ongoing epidemiological studies. It is the opinion of this referee that this manuscript should be published in its current form.
Author Response
Thank you for your comments and appreciation of our work.
Reviewer 4 Report
Reviewer comments for ijerph-2119089
After the Covid19 pandemic began, the scientific community has focused on long and short term goals to stem the current pandemic and prevent future outbreaks. Immuno-surveillance is one such step and needs identifying novel reservoirs of known and unknown viruses of interest to human health. To that end, authors have identified novel Coronaviruses from bat fecal matter and performed phylogenetic analyses on their viromes. They conclude that the beta-coronaviruses found in bats have commonalities with hedgehogs, can spill over into hedgehogs and can result in human infections in the future. While the topic handled here is relevant, the manuscript itself is very poorly written. I recommend rejecting the manuscript and the authors redo their story.
General comments
1)Several instances of incorrect English. I recommend the authors to reach out to either a professional service or a collaborator who is fluent in English.
2)Binomial names of organisms have been written incorrectly and in a very haphazard manner.
3)There are several instances where authors jump topics and make the paragraph hard to follow line 513, 516 for example.
Line by line comments
Introduction
line 95 - Authors have presented a lot of background information but no real statement as to why this study was done. So lots of bats harbor lots of coronaviruses, that can infect other animals and then jump hosts to humans. How exactly does this study add to this body of knowledge? What gap in knowledge is this study addressing?
Lines 213-215 are unnecessary.
Figures - Please do the figures 4 and 5 to make them look bigger. This is the crux of your phylogenetic study and it is impossible to read this tree if I had printed out this manuscript. I have to zoom in and squint even to read the PDF especially in Fig 5.
Figure 6 - Is this the monomer or trimer of the spike protein? Please show the individual monomers/oligomers used for the docking separately. Also, did the docking change when checked for the trimer? Or is that not a normal practice? In real world, a receptor would never interact with a free floating glycoprotein monomer to result in a true infection.
Table 5 - What do the authors make out from finding all these overlapping residues in binding sites?
Line 448 - Some sentence like this should be the hypothesis included in the introduction.
Line 467 - It has taken me to get to the discussion to understand why this study was performed. Authors need to write a better introduction.
Line 473 - Again why is something like this not included in the abstract or introduction?
Line 516 - What were these probabilities? How are they calculated and why are they not listed either the manuscript or in supplementary tables?
Author Response
Thank you for your comments.
Common answers:
- We've fixed languages bugs (corrected by a professional proofreader).
- Binomial names of eucaryotic organisms and names of viruses were corrected. All species names are in italics.
- We have significantly changed and revised the text in the discussion section (including the 513, 516 lines mentioned by the reviewer). The authors hope that the new version of the text is better understandable.
- «Lines 213-215 are unnecessary». We are disagree with reviewer opinion, because lines 213-215 (numbered according to the first version of the text) explain why we assume that all betacoronavirus carrier bats are infected with the same virus.
- The figures of phylogenetic trees have now changed significantly. We hope the new version is much better.
Question: Figure 6 - Is this the monomer or trimer of the spike protein? Please show the individual monomers/oligomers used for the docking separately. Also, did the docking change when checked for the trimer? Or is that not a normal practice? In real world, a receptor would never interact with a free-floating glycoprotein monomer to result in a true infection.
Answer: We have decided that some of the information provided about Structural modeling is redundant. We have removed the figure depicting the protein-protein interaction because it is not informative enough. We plan to conduct experiments in the near future and evaluate the in vitro interaction between spike proteins and animal receptors.
Question: Table 5 - What do the authors make out from finding all these overlapping residues in binding sites?
Answer: We suppose that since there are many overlapping binding sites between Spike-protein of MOW-BatCoV/15-22 and DPP4 of M. brandtii and E. europaeus, the MOW-BatCoV/15-22 is likely can infect both bats and hedgehogs.
Questions: line 95 - Authors have presented a lot of background information but no real statement as to why this study was done. So lots of bats harbor lots of coronaviruses, that can infect other animals and then jump hosts to humans. How exactly does this study add to this body of knowledge? What gap in knowledge is this study addressing?
Line 448 - Some sentence like this should be the hypothesis included in the introduction.
Line 467 - It has taken me to get to the discussion to understand why this study was performed. Authors need to write a better introduction.
Line 473 - Again why is something like this not included in the abstract or introduction?
Answer: I do not think that information about possible (not proven) recombination between viruses as well as the detaled description modern opinion possibility of interspecies transmission should be included namely into the introduction. When we started this study, the main goal was simply to find out whether there are MERS-related coronaviruses in central Russia (this is densely populated area). We found out they are! This information in itself is important enough to be published.
The fact, that these MERS-related viruses may be recombinants is an important observation. The suggestion that hedgehogs may be intermediate hosts is also important. But if we add information about recombination between viruses to the introduction, then it would appear that we were purposefully looking for recombination events. This is not true.
Question: Line 516 - What were these probabilities? How are they calculated and why are they not listed either the manuscript or in supplementary tables?
Answer: It is not entirely clear what was meant. Are you talking about this claim? « Computer molecular docking modeling identified the DPP4 receptors of species that MOW-BatCoV/15-22 spike glycoprotein is likely able to bind to. All of these have overlapping habitats with people, providing opportunity for spillover in a natural setting»
If this claim is questionable, then the calculations are given in the test, see lines 334-347
Round 2
Reviewer 2 Report
Conclusion: The section devoted to the explanation of the results suffers from the same problems revealed so far. Your storyline in the results section (and conclusion) is hard to follow. Moreover, the conclusions reached are really far from what one can infer from the empirical results.
Author Response
- We have changed the logic of the narrative in the "introduction". We hope that now the logic of the introduction is more consistent with the content of the article.
- The "Results" section has also been some changed (at the beginning).
- The "Сonclusions" now they literally correspond to the obtained empirical results.
- Unfortunately, we cannot agree with the proposal to change the Discussion section. This section contains a number of statements of the "hypothesis" category. We believe that the proposed hypotheses follow logically from the results. The presence in the "Discussion" section of a small amount of information about the results obtained in the work is necessary to explain the hypotheses.